# Emergence and Comparative Genome Analysis of *Salmonella* Ohio Strains from Brown Rats, Poultry, and Swine in Hungary

**DOI:** 10.3390/ijms25168820

**Published:** 2024-08-13

**Authors:** Ama Szmolka, Zsuzsanna Sréterné Lancz, Fanni Rapcsák, László Egyed

**Affiliations:** 1HUN-REN Veterinary Medical Research Institute, 1143 Budapest, Hungaryegyed.laszlo@vmri.hun-ren.hu (L.E.); 2National Food Chain Safety Office, 1095 Budapest, Hungary; lanczzs@nebih.gov.hu

**Keywords:** *Salmonella* Ohio, core genome MLST, whole-genome sequencing, food-producing animals, brown rat, poultry, virulence genes, antimicrobial resistance genes, comparison

## Abstract

Rats are particularly important from an epidemiological point of view, because they are regarded as reservoirs for diverse zoonotic pathogens including enteric bacteria. This study is the first to report the emergence of *Salmonella* serovar Ohio in brown rats (*Rattus norvegicus*) and food-producing animals in Hungary. We first reveal the genomic diversity of the strains and their phylogenomic relationships in the context of the international collection of *S*. Ohio genomes. This pathogen was detected in 4.3% (4/92) of rats, captured from multiple sites in Hungary. A whole-genome-based genotype comparison of *S*. Ohio, Infantis, Enteritidis, and Typhimurium strains showed that 76.4% (117/153) of the virulence and antimicrobial resistance genes were conserved among these serovars, and none of the genes were specific to *S*. Ohio. All *S*. Ohio strains lacked virulence and resistance plasmids. The cgMLST phylogenomic comparison highlighted a close genetic relationship between rat and poultry strains of *S*. Ohio from Hungary. These strains clustered together with the international *S*. Ohio genomes from aquatic environments. Overall, this study contributes to our understanding of the epidemiology of *Salmonella* spp. in brown rats and highlights the importance of monitoring to minimize the public health risk of rodent populations. However, further research is needed to understand the route of infection and evolution of this serovar.

## 1. Introduction

*Salmonella* spp. are zoonotic foodborne pathogens included in compulsory annual monitoring, as salmonellosis is still the second most commonly reported zoonotic infection in humans in the EU [1]. *S. enterica*, regarded as the most pathogenic species comprises more than 2600 serovars, of which Enteritidis, Typhimurium, Infantis, and Derby are among the top five *Salmonella* serovars involved in human infections [1,2]. Moreover, there is an increasing number of reports on the emergence of atypical, H_2_S-negative *Salmonella* serovars, which further contributes to the population heterogeneity of *Salmonella* [3,4].

The infection success of this pathogen mostly relies on a number of virulence genes encoding type III secretion systems (T3SS-1 and -2), flagella and on the Salmonella virulence plasmid pSEV, carried by invasive serovars, such as *S*. Enteritidis and Typhimurium [2,5,6]. However, the treatment of salmonellosis has become even more difficult, because some major serovars develop multidrug-resistance through the acquisition of mobile genetic elements such as resistance plasmids, integrons, and transposons. This is of particular importance for *S*. Infantis, where the pESI-like plasmids contributed to the emergence and global spread of this serovar in poultry [7,8,9,10].

The need to understand the virulence and antimicrobial resistance genotypes and to reveal the phylogenetic relationship between the *Salmonella* strains drives the development of robust genotyping strategies. Whole-genome-based approaches, such as core genome multi-locus sequence typing (cgMLST), has therefore become a widely accepted technique for the deep genotyping of several enteric pathogens, including *Salmonella* spp. [11].

The primary sources of human *Salmonella* infections are poultry and poultry products; however, non-typhoidal *Salmonella* has a broad host range, including a multitude of wild or peridomestic animals, including rodents [12,13]. *Rattus* species are particularly important from an epidemiological point of view, as these animals (mostly brown and black rats) live in close proximity to human peridomestic areas and settlements. These species also have close, direct associations with livestock farms, slaughterhouses, and human food waste, and therefore can contribute to the transmission of multidrug-resistant pathogens between animals, humans, and the environment [14,15]. This study aims to characterize the prevalence, serovar composition, and genomic features of *Salmonella* spp. isolated from brown rats derived from different capture sites from Hungary.

Herein, we report the first emergence of the *Salmonella* serovar Ohio in brown rats (*Rattus norvegicus*) and food-producing animals in Hungary. We further provide the whole-genome comparison of the strains, and reveal their phylogenomic relationships with the international *S*. Ohio genomes derived from multiple sources. This study contributes to our understanding of the epidemiology of *Salmonella* spp. in brown rats and highlights the importance of monitoring their living environments to minimize the public health risk related to rodent populations.

## 2. Results

### 2.1. Prevalence and Distribution of S. Ohio Strains in Brown Rats and Food-Producing Animals

*Salmonella* spp. were detected in 4 of the 92 brown rat samples (4.3%), all of which were derived from the same zoological garden (Appendix A and Table 1). Multiple *Salmonella* colonies were isolated from each intestinal sample in parallel to identify potential associations between serovars, however, only strains belonging to the *S*. Ohio serovar (antigenic profile: 6,7:b:l,w) were identified. It should be noted that during the study period, no infection was detected in the Patagonian mara, giant anteater, mantled guereza, or gray parrots, whose enclosures were in close proximity to the capture sites of the rats.

For comparison, the prevalence of *S*. Ohio in some food-producing animals was also determined based on data from the national *Salmonella*-control program (broiler chicken and fattening turkey). *S*. Ohio strains of swine and geese were isolated from carcasses sent for pathological examination to the Veterinary Diagnostic Directorate of National Food Chain Safety Office. *S*. Ohio was very rare in the tested healthy animal flocks of broiler chicken and fattening turkey. In the 2021 survey, *S*. Ohio was not isolated from these animal populations. In 2022, one strain was isolated from broiler chicken and two from fattening turkey flocks (Table 1).

The date collection system was different for swine and geese as these population are not subjected to a national control program. Healthy animals are not routinely tested for *Salmonella,* therefore, the data show the frequency of *S*. Ohio prevalence within all the *Salmonella*-positive samples isolated from carcasses of these animals (Table 1). Two strains were isolated from geese in 2021 and two in 2022. Only one strain was isolated from a swine in 2022 (Table 1). *S*. Ohio was not isolated from any other food-producing animal population in 2021 and 2022.

### 2.2. Phenotypic Features of S. Ohio Strains from Hungary

The 12 *S*. Ohio strains representing the Hungarian basic collection of *S*. Ohio from brown rats and different food-producing animals (Table 1) were tested for antibiotic resistance phenotypes to select representative strains for comparative genome analysis. The results of antimicrobial susceptibility testing showed that all strains were susceptible to the nine clinically relevant antibiotics tested.

Typical black-centered (H_2_S-positive) colonies were detected for all strains on *Salmonella*-selective XLD agar plates. Interestingly however, on Rambach agar plates, we found that *S*. Ohio colonies, especially those from rats, showed a pale pink color instead of crimson, in comparison to the control strains *S*. Enteritidis ATCC 13076, *S*. Typhimurium ATCC 14028, and *S*. Infantis SI54/04 (Figure 1).

### 2.3. Comparison of the Genomic Determinants of Virulence and Antimicrobial Resistance

Five Hungarian *S*. Ohio strains were selected for comparative genome analysis to represent food-producing animals, such as chickens, turkeys, geese, and swine, and their potential food resource competitors, such as brown rats (Table 1). In order to describe the diversity of virulence and antibiotic resistance genotypes and to identify genes that could potentially be unique to this serovar, these strains were analyzed in the context of published complete genomes and of representative *S*. Ohio strains and the most highlighted *Salmonella* serovars of public health significance.

According to the CARD- and VDFB-based in silico analysis of virulence and resistance genotypes, the selected strains of *S*. Ohio and the representatives of the Infantis, Enteritidis, and Typhimurium serovars were comparable on the basis of altogether 153 virulence and antimicrobial resistance genes. Of these, 75 virulence and 42 resistance genes were commonly identified in all the strains and serovars tested (Appendix A), while 29 virulence and 7 antibiotic resistance genes showed differing distributions among the strains (Figure 2). The vast majority (approx. 75%) of the overlapping virulence genes encode key determinants of the SPI-1 (*Salmonella* Pathogenicity Island 1) and SPI-2 Type III Secretion Systems (T3SS-1 and T3SS-2), which play crucial roles in *Salmonella* infection. Accordingly, major gene clusters such as *sip*, *sop*, and *sse* for the T3SS-1,2 effectors and *inv*, *spa*, *prg*, and ssa genes for the assembly and regulation of the needle apparatus were identified in all strains. Similarly, the chromosomal gene clusters *csg* and *fim*, responsible for biofilm formation and host–cell adhesion, were also commonly found. Regarding the comparison of antimicrobial resistance determinants, the genomes of *S*. Ohio, *S*. Infantis, *S*. Enteritidis, and *S.* Typhimurium shared very similar antimicrobial resistance genotypes, especially in relation to genes contributing to chromosomally encoded antibiotic efflux and target-associated resistance mechanisms (Appendix A).

Nevertheless, the results of genotype comparison showed that the T3SS-1 gene *avrA*, the T3SS-2 genes *pipB2*, *sifA*, and *sspH2*, the T3SS-1,2 gene *slrB*, and the fimbrial adhesion genes *lpfABCDE* were present in all strains, except *S*. Ohio. In addition, the plasmid-encoded fimbrial genes *pefABCD*, the immune modulation gene *rck*, and the *Salmonella* plasmid virulence genes *spvBCR* were also not detectable in *Salmonella* serovars Ohio and Infantis (Figure 2). Considering mobile resistance, the extended-spectrum beta-lactamase (ESBL) gene *bla*_CMY-2_ showed the highest prevalence, being identified in four *S*. Ohio strains, including the reference, whereas the Hungarian strains were absent from any plasmid-related antimicrobial resistance determinants (Figure 2).

### 2.4. Phylogenetic Relation of S. Ohio Strains with Different Geographical Distributions

To reveal the phylogenetic relation and genomic diversity, an online collection of *S.* Ohio strains was established based on a targeted search within the NCBI *Salmonella* Genome database. The database search yielded 95 *S*. Ohio strains (isolated between 2000 and 2022), selected to represent a European distribution, but reference genomes from non-European countries were also included (Appendix A). In total, 76.8% of the strains derived from humans and only a small set of *S*. Ohio strains from livestock and the environment were available for genome comparison to the five Hungarian strains, representing different sources from poultry, swine, and intestinal samples from brown rats.

The sequence type (ST) of the strains was identified using MLST, based on the available whole-genome sequences. The Hungarian strains 512/S1 (brown rat), 194/21 (goose), 453/22 (swine), SM-230/22 (broiler chicken), and SM-804/22 (fattening turkey) were identified as ST329, the sequence type where 96.8% of the compared *S*. Ohio genomes were allocated (Appendix A).

To provide a deeper understanding of genomic diversity, a cgMLST analysis of the *S*. Ohio strains was performed based on the polymorphism of 3531 genes of the core genome by using the strain *S*. Ohio- SA20120345 as a reference. The core genome phylogenetic tree displayed substantial diversity among the compared genomes of *S*. Ohio from animals and humans, with a global distribution (Figure 3). Overall, four clusters and numerous subgroups were distinguished, of which Cluster 1 exhibited the most significant genomic diversity. This cluster comprised 54.8% of the human *S*. Ohio strains and all strains originating from swine (including 453/22), except CVM N44711 from the U.S. Hungarian strains from rat and poultry were grouped in Cluster 2 together with four published *S*. Ohio strains from river water and human, bovine, and poultry isolates from Mexico and the UK, respectively. Within Cluster 2, the Hungarian strains were closely grouped together in a subcluster, in which the rat and goose strains shared the highest cgMLST genotype identity (Figure 3).

## 3. Discussion

*Salmonella* remains the second most prevalent zoonotic pathogen with One Health relevance, and food-producing animals, especially poultry, are the major source of the top five EU-level *Salmonella* serovars posing a continuous pressure on food safety and public health [1]. Genetic flexibility plays a crucial role in the dynamics of *Salmonella* prevalence. This may allow for adaptation, leading to the emergence and spread of currently less significant serovars under certain conditions.

We suppose that brown rats circulating between a broad range of habitats could actively contribute to this emergence by hosting *Salmonella* serovars other than those widespread in livestock and humans. Based on this, the present study aimed to provide the first insight into the prevalence and genomic features of *Salmonella* spp. in brown rats in Hungary.

Wild rodents are regarded as reservoirs for diverse zoonotic pathogens including enteric bacteria. In this study, brown rats from different environments (zoological gardens, animal parks, animal farms, and backyard environments) in Hungary were tested for *Salmonella* prevalence. This pathogen was detected in 4.3% of the screened rats, all of which were captured from the same zoological garden. This low prevalence was comparable to that (0.5–1.4%) found in urban rats [16,17,18]. In contrast, an increased prevalence (10–49%) of *Salmonella* spp. has been observed in rats living in livestock farms and wet market areas, probably because of their close contact with animal feed and meat products [12,14,19]. These results suggest that rats do not represent a *Salmonella* hazard in Hungary, but that the prevalence of *Salmonella* in rats varies greatly among the environments they inhabit. Notably, no *Salmonella* infections were detected in the other animal species that were sampled, despite their close proximity to the rat capture sites. Susceptibility to *Salmonella* infection among different animal species is influenced by the complex interplay between host genetics, microbiota composition, and bacterial diversity. Studies have shown that genetically similar mice from different vendors exhibit marked phenotypic variation in susceptibility to *Salmonella* infection due to changes in the composition of the gut microbiota [20]. Additionally, genetic variants, such as a loss-of-function mutation in the *Itgal* gene, and the presence of diverse *Salmonella* phenotypes expressing different pathogenic traits influence susceptibility to infection in both mice and farm animals [21,22].

Rats can host a wide range of *Salmonella* serovars, some of which are relevant to human infections. Accordingly, in the above-mentioned studies, serovars Enteritidis and Typhimurium were detected in wild urban rats [16,17]; Enteritidis, Typhimurium, Infantis, Corvallis, Mbandaka, Newport, and Anatum in rats from the animal production environment [12,14,23,24]; and Weltevreden in wet markets in Thailand [19]. However, our study is the first to report the presence of the *Salmonella* serovar Ohio in rats and food-producing animals in Hungary. These animals showed an overall low prevalence of *S*. Ohio, which confirms the relevant data from the monitoring reports with a prevalence of 1.3% in laying hens [1,25]. However, local differences may exist, and a study on *Salmonella* bacteriophage diversity predicted a high proportion of *S*. Ohio in broiler and layer farms in Spain [26].

*S*. Ohio is more frequently reported outside Europe, being recovered mostly from poultry and pig production environments [27,28,29]. With all that, *S*. Ohio is regarded as a rare serovar of *Salmonella*, but its occurrence also includes wild animal species and aquatic environments [30,31], which was first described by Joseph et al., (1984) in rodents from Malaysia [32]. Although a few studies have linked this serovar to local human infections [33,34], these data indicate that *S*. Ohio does not yet belong to the *Salmonella* serovars, representing a major risk for food safety and public health in the European region. However, the detection of *Salmonella* spp. in brown rats highlights the need for proper sanitation and hygiene measures in areas where these animals are present.

The production of hydrogen sulfide (H_2_S) and the utilization of propylene glycol (1,2-propanediol) are biochemical properties that hallmark *Salmonella* detection on selective media, such as XLD and Rambach agar. On these selective plates, typical *Salmonella* spp. colonies—except for *S*. Typhi and Paratyphi which fail to utilize propylene glycol—appear as black-centered and crimson, respectively.

Here, we found that all *S*. Ohio strains were H_2_S-positive and produced typical black-centered colonies on XLD plates, whereas rat strains displayed an atypical colony phenotype on Rambach agar plates, which is a novel finding. Gruenewald et al. (1991) indicates that while 98.2% of the *Salmonella*, including Ohio, can grow on Rambach (propylene glycol-containing) agar, some serovars such as Enteritidis, Hadar, and Johannesburg do not produce the typical crimson phenotype, but they are pink or colorless [35]. Strains with atypical colony phenotypes have not yet been described in relation to *S*. Ohio. The pale phenotype on Rambach agar can be attributed to alterations in the metabolic pathway of propylene glycol, which is used as a carbon source in this medium. One possible mechanism is the presence of mutations in propanediol utilization *pdu* genes, which encode enzymes involved in propylene glycol catabolism [35]. Mutations in these genes can lead to a loss or reduction in the enzymatic activity necessary for the conversion of propylene glycol to its metabolites, resulting in the absence of the characteristic crimson phenotype. The inability of some *Salmonella* strains to use propylene glycol as a carbon and energy source could be a disadvantage in competition with typical strains of *Salmonella*, but it could also be an indication of host adaptation. In any case, it could have implications for the development and use of specific diagnostic techniques. One limitation of these findings could be the small number of *S*. Ohio detected in rats, and thus the results may not be generalizable to all *S*. Ohio strains carried by these rodents. Further studies are needed to investigate the prevalence and significance of the atypical colony phenotype of *S*. Ohio strains and its molecular background.

An extensive analysis of non-typhoidal *Salmonella* strains from a comprehensive whole-genome perspective contributes to the identification of unique genetic markers for serovar detection and uncovers the mechanisms underlying potential changes in serovar composition. Here, we provide the first evidence of genomic differences between *S*. Ohio and reference strains of globally spread *Salmonella* serovars, including Enteritidis, Typhimurium, and Infantis, with respect to antibiotic resistance and virulence. This study is one of the few to focus on whole-genome comparisons of virulence and antimicrobial resistance genotypes of diverse *Salmonella* serovars relevant to food production and other environments [36,37,38,39].

Our results are most comparable to those reported by Gupta et al. (2019), who described the pangenome of 14 *Salmonella* serovars derived from swine from the U.S., including one strain of *S*. Ohio [38]. Consistent with the outcomes of these studies, our comparative analysis showed that 76.4% (117/153) of the genomic determinants of virulence and antibiotic resistance were conserved among *Salmonella* serovars Ohio, Infantis, Enteritidis, and Typhimurium. Therefore, these genes are not suitable to distinguish between *Salmonella* strains and serovars. None of the virulence and antibiotic resistance genes displayed specificity for *Salmonella* serovar Ohio.

However, a small subset of virulence genes involved in *Salmonella* infection, such as *avrA* (T3SS-1) *pipB2*, *sifA*, *sspH2* (T3SS-2), and *slrP* (T3SS-1/2), and the fimbrial operon lpf were exclusively detected in serovars Typhimurium, Enteritidis, and Infantis. The absence of *avrA* and *lpfABCDE* in *S*. Ohio was also confirmed by Gupta et al. (2019) [38]. However, there is a contradiction regarding the genes *pipB2*, *sifA*, and *slrP*, which the above-mentioned study reported as being present in all serovars, including Ohio. One reason for these contradictory findings may be the different approaches used for genome comparison and for the in silico detection of genomic determinants of virulence and antimicrobial resistance. The downregulation of SPI-1 genes and a consequent reduction in the pathogenic potential were demonstrated for *S*. Infantis, when compared to *S*. Typhimurium, despite the high genetic similarity considering chromosomal markers for virulence between the two serovars [40]. This supports our hypothesis that an incomplete T3SS repertoire and the absence of key virulence plasmids are associated with reduced pathogenicity, which may explain the overall negligible prevalence of *S*. Ohio strains, but further in vitro/in vivo studies are needed to test the power of this suggestion.

Specific virulence plasmids associated with the invasive phenotype of *Salmonella* [6] and the pESI-like hybrid plasmids (resistance/virulence) described for the global clones of *S*. Infantis [7,8,9,10] were also not characteristic of the *S*. Ohio strains. Despite the presence of genes encoding multidrug efflux systems, we found that *S*. Ohio strains from Hungary were susceptible to all antibiotics tested, which was also confirmed by the absence of antimicrobial resistance plasmids. This finding suggests that *S*. Ohio is a new, recently imported/entered serovar in Hungary, and therefore not yet “adapted” to local farming and environmental conditions, including antibiotic administrations.

In contrast, some *S*. Ohio strains isolated from food-producing animals have been described as multiresistant, with transferable plasmids conferring resistance to phenicols (*floR*), aminoglycosides (*aph(3″)* and *aph(6)*), beta-lactams (*bla*_CMY-2_), tetracyclines (*tet*(A)), and sulfonamides (*sul1* and *sul2*) [28,38,41]. This is not surprising, because of the horizontal transfer of mobile genetic elements, these antimicrobial resistance genes and plasmids have become widespread among enteric bacteria recovered from both livestock and humans. Extended-spectrum-β-lactamase (ESBL) genes are considered a priority group of genes from a food safety point of view; therefore, *bla*_CMY-2_-positive *S*. Ohio strains pose a possible risk to food safety and public health.

Among the whole-genome-based techniques, cgMLST is recognized as a robust tool for modern epidemiology to reveal the genomic diversity of bacteria, including *Salmonella* spp. Here, we used cgMLST phylogeny to reveal the phylogenomic relationships between the Hungarian strains of *S*. Ohio from rats, poultry, and swine in the context of international *S*. Ohio genomes derived from diverse sources. Previous genome-based studies on *Salmonella* also focused on phylogenomic relationships between outbreak strains [42] or within certain emerging clones, such as the PFGE clone B2 of the multiresistant strains of *S*. Infantis [43]. This study is the first to describe the cgMLST-based genotype diversity of *Salmonella* serovar Ohio in Hungary, in relation to international strains of this serovar.

The phylogenomic comparison highlighted the close genetic relatedness between the rat and poultry strains of *S*. Ohio from Hungary. These strains clustered closely together with a set of *S*. Ohio isolates derived from aquatic environments [30]. This genetic constellation may be of particular importance in relation to rat strains, isolated in the zoological garden of Pécs. This zoo was famous for being the only rat-free zoo in Hungary. However, due to construction in the spring of 2022, the sewer system had to be opened. This was an opportunity for the rats to take advantage of and enter the zoo area through the canal. Rats are considered “sponges” that reflect their living environments rather than primary hosts of *Salmonella* [23]. Therefore, this scenario could explain why *S*. Ohio strains from rats and those from aquatic environments are closely related. This indicates that sewage water can be a reservoir for rare and exotic *Salmonella* serovars, and that *Rattus* species preferably populating this environment may contribute to the emergence of new pathogens with future significance from the perspective of OneHealth.

## 4. Materials and Methods

### 4.1. Samples Collected from Brown Rats

For these studies, a total of 92 free-ranging brown rat (*Rattus norvegicus*) individuals were live-captured in special trap cages at eight sampling sites in Hungary, representing different collection sites: zoological gardens, animal parks, animal farms, and granary and backyard environments (Appendix A). Among them, thirteen brown rats were captured in the zoological garden of Pécs. All individuals were clinically healthy and active. Their capture site was in close proximity to the location where the Patagonian mara (*Dolichotis patagonum*), giant anteater (*Myrmecophaga tridactyla*), mantled guereza (*Colobus guereza*), and gray parrots (*Psittacus erithacus*) were kept.

All captured animals were anesthetized using xylazine HCl (Sedaxylan injection, EUROVET ANIMAL HEALTH B.V., Bladel, The Netherlands) and tiletamine HCl-zolazepam (Zoletil 100 injection, Virbac, Carros, France) and were placed in a CO_2_ atmosphere. As a pest species, brown rats can be killed without permission in Hungary, if they are not kept alive until execution. The large intestines were then aseptically removed and stored at −20 °C until processing. In Hungary, permissions for animal experiments are not required for pest animals if they are not kept beyond the time of sampling and do not undergo any medical treatment. This is stipulated in Paragraph 4 of Law No. 28/1998, accepted by the Hungarian Parliament, and Edict No. 2013/40 of the Hungarian Government.

### 4.2. Sample Processing, Detection, and Identification of Salmonella Strains

Each intestinal sample was used to isolate multiple strains (eight *Salmonella* colonies/sample) of *Salmonella* spp. to represent possible serovar diversity within the positive samples. *Salmonella* was detected according to the international *Salmonella* standard ISO 6579-1:2017 [44] with minor modifications. Briefly, the intestinal contents were diluted in 10× LB broth and incubated for 18 h at 37 °C. Selective enrichment was performed in 10 mL RV (Rappaport-Vassiliadis) broth (Merck Life Science, Burlington, MA, USA) inoculated with 100 µL of overnight LB culture and incubated for 24 h at 42 °C. Next, 10-10 µL of the overnight RV cultures was streaked onto Xylose Lysine Deoxycholate (XLD) and Chromocult^®^ RAMBACH™ agar plates (Merck Life Science, Burlington, MA, USA). To compare colony morphologies, a representative strain of each of the three most prevalent *Salmonella* serovars in Hungary (*S*. Enteritidis, *S*. Typhimurium, and *S*. Infantis) were also included (Figure 1).

After 27 h of incubation (37 °C), *Salmonella*-like colonies were randomly selected and subjected to PCR for species confirmation with *Salmonella*-specific primers, as described by Aabo et al. (1973) [45]. After molecular identification, *Salmonella* isolates were stored at −80 °C in LB containing 10% glycerol.

Serotyping was performed according to international standard ISO/TR 6579-3:2014 [46] with the slide agglutination method based on the White–Kauffmann–Le Minor scheme. Commercial products of group specific and monovalent antisera were used (BioRad Laboratories, Hercules, CA, USA). We established a basic comparative collection of 12 *S*. Ohio strains isolated from the intestinal samples of brown rats (n = 4) and from different food-producing animals (n = 8), provided by the National Food Chain Safety Office (Budapest, Hungary) (Appendix A and Table 1).

### 4.3. Antibiotic Susceptibility Testing of Salmonella Ohio Strains and Selection for Genome Analysis

The antibiotic resistance phenotype of the strains was determined by means of disk diffusion (Kirby–Bauer) against the following nine antibiotic agents: ampicillin (10 µg), cefotaxime (5 µg), chloramphenicol (30 µg), ciprofloxacin (5 µg), gentamicin (10 µg), meropenem (10 µg) nalidixic acid (30 µg), sulphonamide compounds (300 µg), tetracycline (30 µg), and trimethoprim (5 µg). Antibiotic susceptibility testing was performed in accordance with the guidelines and interpretation standards of the European Committee on Antimicrobial Susceptibility Testing [47]. Strains with intermediate zone diameters were considered to be susceptible. *E. coli* strain ATCC 25922 was used as a reference. Five of the above twelve strains of *S*. Ohio were selected for WGS-based comparative genome analysis, to represent each of the host species: brown rat, broiler chicken, fattening turkey, goose, and swine (Table 1).

### 4.4. Whole-Genome Sequencing and Prediction of the Antibiotic Resistance and Virulence Genotypes

Whole-genome sequencing was performed by Eurofins Biomi Ltd. (Gödöllő, Hungary). Short reads of 2 × 250 bp were generated on the Illumina MiSeq sequencing platform using v2 sequencing chemistry. The quality of the raw sequences was analyzed using FastQC v0.12.1 [48], and raw reads were trimmed using the built-in application fastp v0.22.0 of the BIONUMERICS software (bioMérieux SA, Marcy-l’Étoile, France). A de novo genome assembly of the processed reads was carried out with SPAdes v3.15.3 [49]. The genomic contig sequences of the five *S*. Ohio strains studied here were deposited in BioProject PRJNA1049837, with BioSample identifiers listed in Appendix A.

The *S*. Ohio serovar was confirmed in silico based on assembled contig sequences using the web-based tool SeqSero v1.2 [50]. Accordingly, the antigenic profile 6,7:b:l,w (O:H1:H2) was predicted for the five selected *Salmonella* strains. The gene annotation and contig analysis of the five *S*. Ohio strains were performed using the genome analysis toolkit of the Bacterial and Viral Bioinformatics Resource Center (BV-BRC) (https://www.bv-brc.org/; accessed on 20 December 2023). Accordingly, the Virulence Factor Database (VFDB) and Comprehensive Antibiotic Resistance Database (CARD) provided reference data for the in silico prediction of antimicrobial resistance and virulence genes. A threshold of >90% query coverage was applied to detect virulence and antibiotic resistance genes.

### 4.5. Phylogenetic Analysis: MLST and Core Genome MLST

To compare genomic diversity and reveal phylogenetic relationships within the Ohio serovar, a cluster analysis of an online collection of *S*. Ohio genomes was performed based on the core genome genes. For this, a data package of selected *S*. Ohio genomes from humans and multiple animal sources was downloaded from the NCBI *Salmonella* Genome database (https://www.ncbi.nlm.nih.gov/datasets/taxonomy/28901/, accessed on 6 March 2024) (Appendix A).

Phylogenetic analysis was performed using Ridom SeqSphere+ software v9.08 (Münster, Germany) [51]. The core gene set of the tested *S*. Ohio strains was established by a BLAST search against the *S*. Ohio reference strain SA20120345 (GenBank accession: NC_CP030024.1). Sequence types (STs) were determined by multilocus sequence typing (MLST) according to the Achtman MLST scheme for *S. enterica* [52].

## 5. Conclusions

This study is the first to report the emergence of *Salmonella* serovar Ohio in brown rats (*Rattus norvegicus*) and food-producing animals in Hungary. Our findings provide valuable insights into the genomic diversity of the *S*. Ohio strains and their phylogenomic relationships in the context of the international collection of *S*. Ohio genomes. Overall, this study contributes to our understanding of the epidemiology of *Salmonella* spp. in brown rats and highlights the importance of monitoring their living environments to minimize the public health risk of rodent populations. However, further research is needed to understand the route of infection and evolution with possible host adaptation of this serovar.

## Figures and Tables

**Figure 1 ijms-25-08820-f001:**
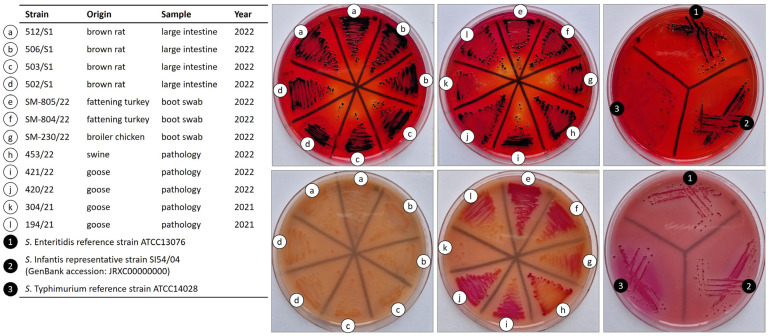
Colony color of *S*. Ohio strains on XLD and Rambach agar plates.

**Figure 2 ijms-25-08820-f002:**
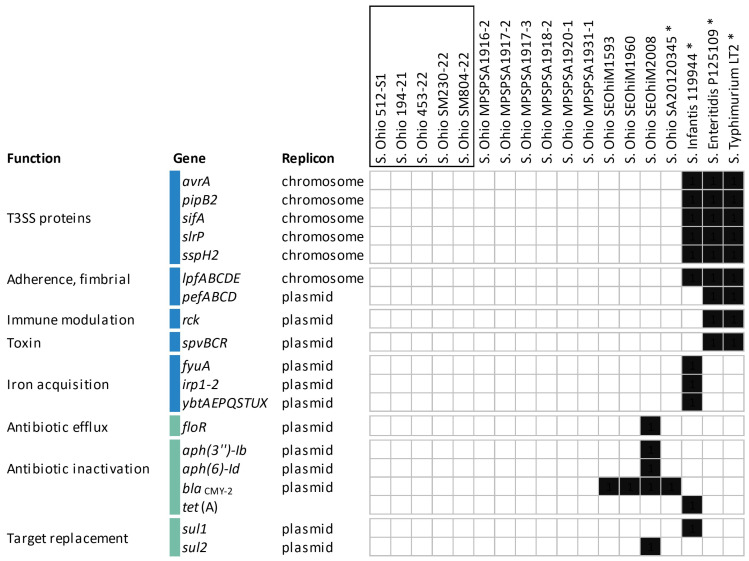
Virulence and antibiotic resistance genes with differential distributions among compared strains of *S*. Ohio, *S*. Infantis, *S*. Enteritidis, and *S*. Typhimurium. A threshold of >90% query coverage was applied to detect the chromosomal virulence and antibiotic resistance genes. The Hungarian strains of *S*. Ohio are shown in frames, and the reference strains are labeled with asterisks. Blue bars indicate virulence genes, whereas green bars indicate antibiotic resistance genes. The BioSample numbers of the published *S*. Ohio strains can be found in Appendix A, while the reference strains *S*. Ohio SA20120345, *S*. Infantis 119944, *S*. Enteritidis P125109, and *S*. Typhimurium LT2 are referred under SAMN09237646, SAMN13850655, SAMN16552335, and SAMN02604315, respectively.

**Figure 3 ijms-25-08820-f003:**
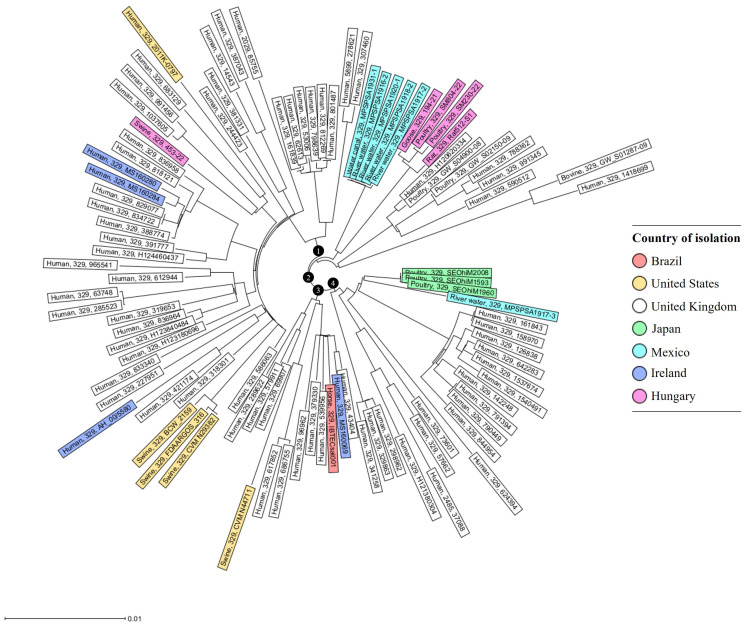
Core genome diversity and geographical distribution of *S*. Ohio strains isolated from multiple animal, human, and environmental sources. The Neighbor Joining Tree showing the genomic diversity and phylogenetic relation of 95 *S*. Ohio strains was calculated based on the polymorphism of 3531 target genes of the core genome. Core genes were identified by blasting all genome sequences against the *S*. Ohio reference strain SA20120345 (GenBank accession: NC_CP030024.1). The symbols ❶–❹ indicate the cgMLST clusters.

**Table 1 ijms-25-08820-t001:** Prevalence of *S*. Ohio strains in samples from rats and food-producing animals in Hungary.

Animal/Source	Total Tested	Prevalence (n)	Strains
Brown rat	n = 92	4.3% (4)	502/S1; 503/S1; 506/S1; 512/S1 *
Broiler chicken (2022) ^a^	n = 17,134	6‰ (1)	SM-230/22 *
Fattening turkey (2022) ^a^	n = 3593	0.056% (2)	SM-804/22 *; SM-805/22
Goose (2021) ^b^	n = 3593	0.056% (2)	194/21 *; 304/21;
Goose (2022) ^b^	n = 130	1.5% (2)	420/22; 421/22
Swine (2022) ^b^	n = 48	2.0% (1)	453/22 *

^a^ National *Salmonella*-control program; ^b^ Pathological sample; * Strains selected for comparative genome analysis.

## Data Availability

The datasets generated for this study can be found in the NCBI database BioProject PRJNA1049837.

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
