# Peer review of "Emergence and Comparative Genome Analysis of Salmonella Ohio Strains from Brown Rats, Poultry, and Swine in Hungary"

_ijms, 2024, doi:10.3390/ijms25168820_

Round 1

Reviewer 1 Report

Comments and Suggestions for Authors

The manuscript reported the first emergence of the Salmonella serovar Ohio in brown rats and food-producing animals in Hungary. It further provided a whole-genome comparison of the strains, revealing their phylogenomic relationships with international S. Ohio genomes derived from multiple sources. This text contributes to our understanding of the epidemiology of Salmonella spp. in brown rats and highlights the importance of monitoring their living environments to minimize public health risks from rodent populations.

However, the explanation of the correlation revealed in Figure 3 is inadequate, particularly regarding Cluster 2. Additionally, the manuscript did not investigate how the strains are transmitted from wild animals to humans, and the number of strains studied is too small with a regional distribution that may not represent the entire area of Hungary. Therefore, further research is needed to understand the routes of infection and evolution of this serovar in the future.

Specific suggestions are as follows:

Title: For greater clarity, it is recommended to use: “Emergence and Comparative Genome Analysis of Salmonella Ohio Strains from Brown Rats, Poultry, and Swine in Hungary”.

Page 2, Line 68-70: The total number of rats and the selected area are insufficient; the data are too limited to support the relevant discussion, making it difficult to draw exact conclusions.

Page 2, Line 85-86: The bacteria isolated from the carcass may not be originated from the animal itself but could have been acquired post-mortem. How can it be ruled out that the bacteria isolated from the body were not post-mortem contaminants?

Page 3, Line 96-97: Further evidence suggests that the bacteria in the bodies may have originated externally. Please provide additional details.

Page 3, Line 106: Please explain the mechanism behind the observed color change.

Page 5, Line 170: The symbols ①②③④ in the graph should be annotated.

Page 6, Lines 212-213: The discussion here would benefit from further theoretical support, such as differences in receptors in epithelial cells.

Page 8, Line 328: An animal ethics permit should be attached here.

Page 10, Line 414: How feasible is it to regulate the living environment of wild animals? Please provide practical recommendations.

Author Response

We would like to thank the reviewer for their critical review and favourable evaluation of the manuscript. Most of the reviewers’ suggestions are related to Section 2.1. of the results, where the representation of the brown rats and the isolation of the S. Ohio strains caused some uncertainty and required clarification.

Considering the limited number of rats in this study, we have to note that this is not an epidemiological study, but rather an extended case report about a rat population of a Zoopark that was found positive for a Salmonella serovar that was previously not detected in Hungary.

Sampling of animal carcasses in the frame of pathological examination was performed using sterile devices, therefore, the possibility of cross-contamination can be excluded. However, it cannot be determined whether S. Ohio was present in the animal as a part of the microbiome or could lead to the illness of the animal.

A potential explanation for the mechanism underlying the observed pale phenotype of Salmonella on Rambach agar plates was included in the corresponding part of the discussion section (L. 250-256).

The legend of Figure 3 has been completed as indicated (L. 173-174).

The discussion on page 6 was extended to include information on the mechanisms underlying the differences in susceptibility to Salmonella infection (L.211-218). Consequently, three new references were been added to the reference list.

We have included the requested information regarding animal ethics in L.351-355.

Considering the last question of the reviewer on page 10, we would suggest that to monitor the living environment of wild animals, technologies such as camera traps, GPS collars and drones can be used to gather data on wildlife (rodent) movement and behaviour.

We hope that, by incorporating the requested information, the new revised (R1) version of the manuscript will now be suitable for publication.

Reviewer 2 Report

Comments and Suggestions for Authors

Please revise your manuscript with comments provided in the attached manuscript.

Comments on the Quality of English Language

Author Response

We thank the reviewer for the critical review and for the favourable evaluation of the manuscript. All sections have been revised and corrected as requested, according to the reviewers’ comments and suggestions.

Concerning the reviewer’s first question (“Why did not introduced to S. Ohio health significance in your region”), the public health significance of S. Ohio was not included in the Introduction because to our knowledge, there are no previous data related to the occurrence and potential public health relevance of this Salmonella serovar in this region and Hungary.

L.73-75.: Because the capture site of the brown rats was in close proximity to the location where the Patagonian mara, giant anteater, mantled guereza and gray parrots were kept, these animals were also tested for the presence of Salmonella spp., but none of them were infected with this pathogen.

The discussion on page 6 was extended to include information on the mechanisms underlying the differences in susceptibility to Salmonella infection (L.211-218). Consequently, three new references were been added to the reference list.

We hope that, by incorporating the requested information, the new revised (R1) version of the manuscript will now be suitable for publication.

Reviewer 3 Report

Comments and Suggestions for Authors

The authors in the present study characterize  S. Ohio isolates from rats and food producing animals in Hungary. The number of the studied isolates is very small. However, the scientific approach of the subject from the authors makes the manuscript worth to be published.

Some revisions need to be addressed:

- the English language in some parts of the manuscript should be revised. I suggest the text to be thoroughly  checked by an English speaker.

Line 32-34: OK. not so important to be included in the introduction. 

Line 40: Salmonella's resistance to antibiotics is attributed to the presence of mobile genetic element: plasmids, integrons, transposons

Line 59: the fact that the authors detected S.Ohio for the first time in brown rats , it does not mean that it was emerged in brown rats

Line 61: “We further provided the 60 whole-genome based comparison of the strains…”

Line 65: health risk related to rodent populations

Line 94: 2-2?

Line 105: XLD agar

Lines 193-195: Please rephrase. It is not clear what the authors want to say.

Lines 293-295: I am not sure that the fact that few (5?) S. Ohio isolates were pan-susceptible leads to the conclusion that they are recently emerged. There are several Salmonella serotypes circulating in EU that are also pan-susceptible.

Line 319: construction? Or re-construction? Please rephrase “must be opened” in terms of better use of the English language

Line 332: 11 rats or 13 (Table S1)? The first 2 of the Table Si were excluded due to their weight?

Lines 367: Disk diffusion method (Kirby-Bauer)

Line 350: 10-10?

Line 389: 6,7:b:l,w is the full antigenic type

Comments on the Quality of English Language

- the English language in some parts of the manuscript should be revised. I suggest the text to be thoroughly  checked by an English speaker.

Author Response

We thank the reviewer for the critical review and for the favourable evaluation of the manuscript. All sections have been revised and corrected as requested, according to the reviewers’ comments and suggestions.

We hope that, by incorporating the requested information, the new revised (R1) version of the manuscript will now be suitable for publication.
